

# Molecular characterization of nearshore baitfish populations in Bermuda to inform management

Gretchen Goodbody-Gringley[1], Emma Strand[2,3] and Joanna M. Pitt[4]

[1] Bermuda Institute of Ocean Sciences, St. George, Bermuda
[2] University of Rhode Island, Kingston, RI, United States of America
[3] Loyola Marymount University, Los Angeles, CA, United States of America
[4] Department of Environment and Natural Resources, Bermuda Government, Bermuda

Corresponding author
Gretchen Goodbody-Gringley,
gretchen.ggringley@bios.edu

## ABSTRACT

Small-bodied marine fishes play an important role in the food web, feeding both larger fishes and seabirds. Often referred to as baitfishes, they concentrate seasonally in coastal areas in large, often heterospecific assemblages that are targeted by both commercial and recreational fishers. Given apparent declines in at least some of Bermuda's baitfish species over the past 40 years, it is useful to determine the species composition of baitfish assemblages, and how it varies among sites, in order to inform management. Using genetic barcoding of the Cytochrome c oxidase 1 gene (COI), we confirm species identity, assess intraspecific genetic diversity locally, and determine rates of broader genetic connectivity for baitfish assemblages in Bermuda. Species analyzed included *Hypoatherina harringtonensis*, *Anchoa choerostoma*, *Jenkinsia lamprotaenia*, *Harengula humeralis*, *Opisthonema oglinum* and *Sardinella aurita*. Species identification based on molecular barcoding revealed some misidentification of individuals based solely on gross morphological characteristics, with an error rate of 11%, validating the usefulness of this approach. Interestingly, sequence results for the endemic Bermuda anchovy, *A. choerostoma*, were within 1% similarity to the more broadly distributed big-eye anchovy, *A. lamprotaenia*, and thus additional analyses are warranted to evaluate the genetic basis for endemism. Estimates of genetic diversity within and among baitfish assemblages in Bermuda were high, indicating high rates of local connectivity among sites for all species. As such, management should consider Bermuda's baitfish species as single, highly mixed populations. However, with the exception of *H. humeralis* and the endemic *A. choerostoma*, significant genetic differentiation and population structure were found when comparing Bermuda's baitfish populations with those from other regions, suggesting limited gene flow between other regions and Bermuda for these species. Limited regional connectivity has implications for management, as strong genetic divergence suggests that populations in Bermuda are predominantly self-seeding and thus not likely to be replenished from distant populations. These results therefore support precautionary management of baitfish species in Bermuda.

## INTRODUCTION

Small-bodied, shoaling marine fishes are a critical part of the food chain, connecting plankton at low trophic levels to higher trophic level organisms such as seabirds and piscivorous fishes (*Smith et al., 2011*; *Pikitch et al., 2014*). These species form large aggregations in coastal areas, and here they are targeted by both commercial and recreational fishers (*Smith-Vaniz, Collette & Luckhurst, 1999*; *Smith et al., 2011*). Commonly referred to as 'forage fish' or 'baitfish', their combined ecological and fisheries importance makes these species a priority for management (*Smith et al., 2011*; *Pikitch et al., 2014*).

In Bermuda, large, heterospecific baitfish aggregations typically include several morphologically similar species from the families Clupeidae, Engraulidae, Atherinidae, and Hemiramphidae (*Parrish, 1989*; *Smith-Vaniz, Collette & Luckhurst, 1999*). Species frequently targeted by fishermen include the Reef silverside, *Hypoatherina harringtonensis* (G.B. Goode, 1877) [Family Atherinidae], the endemic Bermuda anchovy, *Anchoa choerostoma* (G.B. Goode, 1874) [F. Engraulidae], and the Dwarf herring, *Jenkinsia lamprotaenia* (P.H. Gosse, 1851) [F. Dussumieriidae], as well as the larger and morphologically similar Redear herring, *Harengula humeralis* (G. Cuvier, 1829), Round sardinella, *Sardinella aurita* (A. Valenciennes, 1847), and Threadfin herring, *Opisthonema oglinum* (C.A. Lesueur, 1818), all [F. Clupeidae] (*Smith-Vaniz, Collette & Luckhurst, 1999*; *Lavoué, Konstantinidis & Chen, 2014*).

Targeted commercial baitfish fisheries in Bermuda utilize seine nets to harvest shoaling fish, which are then used secondarily as either line bait or chum in further fishing activities, or sold to recreational fishers. Recreational fishers may use a cast net only to catch bait for personal use (Bermuda Fisheries Regulations 2010). Prior to the banning of fish traps in 1990, these small, oily fishes were also placed in mesh bags that were added to traps in order to draw larger fishes to them from greater distances (*Butler et al., 1993*; *Smith-Vaniz, Collette & Luckhurst, 1999*). The annual harvest of baitfishes in Bermuda peaked at 105,072 kg in 1988, corresponding with the peak of local trap fishing activity, and fell to a low of 26,842 kg in 1995 (*Butler et al., 1993*; *Smith-Vaniz, Collette & Luckhurst, 1999*). Long-term landings statistics show that, after a decade of adjustment, total reported commercial catches of baitfish species have remained largely stable in the range of 30,000–40,000 kg per annum between 1999 and 2017 (Bermuda Government Department of Environment and Natural Resources, pers. comm., 2018 http://www.environment.bm). The Dwarf herring and Bermuda anchovy, which, along with the Reef silverside are collectively called 'fry', make up the bulk of the baitfish harvest (*Smith-Vaniz, Collette & Luckhurst, 1999*). However, anecdotal evidence from fishers, together with landings records for the larger baitfish species, suggests that local populations of at least some species have declined over the past 40 years.

For species targeted by fishing, overexploitation can lead to population declines (*Ecoutin et al., 2010*; *Last et al., 2010*; *Stagličić et al., 2011*). Other anthropogenic impacts in coastal areas, such as pollution and habitat degradation (*Kennish, 2002*; *Hewitt et al., 2008*; *Johnston & Roberts, 2009*), can also alter fish distribution patterns, decreasing richness and abundance across various spatial and temporal scales (*Sax & Gaines, 2003*; *Johnston &*

*Roberts, 2009*). Apparent declines in the abundance of baitfishes in Bermuda may therefore reflect natural fluctuations in the abundance and distribution of these species, or may be indicative of overfishing or other anthropogenic impacts. However, a significant change in fishing practices, such as the banning of fish traps, may also affect the ways in which a related resource, such as bait, is harvested and used, which may in turn affect how that resource is perceived and monitored by fishers.

At present, regulation of baitfishing in Bermuda under the Fisheries Act, 1972, and the Fisheries Regulations, 2010, includes both gear and spatial restrictions. In particular, there are four inshore bays (Somerset Long Bay, Shelly Bay, Whalebone Bay, and Coot Pond), within which the use of fishing nets and the removal of baitfishes is completely prohibited (Bermuda Fisheries Act, 1972, Section 8i), but these bays are not evenly distributed around the island (Fig. 1). Further, given the similarities in species morphologies, visual identification of baitfish species in the field can be difficult, and a higher diversity may exist within baitfish schools than is currently perceived. Thus, increased understanding of how assemblage composition varies across locations, along with the extent of local movements and genetic mixing, is required to inform management.

Lastly, if Bermuda's baitfish populations are indeed experiencing declines, a greater understanding of regional genetic connectivity could indicate whether or not larval supply from other populations might assist with their recovery (see *Cowen & Sponaugle, 2009*). Bermuda's isolated, mid-Atlantic location (Fig. 1, inset) reduces the likelihood of regular supply of larvae from external sources (*Schultz & Cowen, 1994*), and there is evidence indicating that at least some local fish populations are self-seeding (see *Locke et al., 2013*). However, despite this isolation, Bermuda was the first location outside of the eastern coast of the United States where invasive lionfish were detected (*Whitfield et al., 2002*), indicating that larval transport and/or post-larval rafting in association with floating material across this distance is not only possible but occurs at a rate that enabled successful establishment of an invasive species (*Locke et al., 2013*). Further, Bermuda has low rates of endemism (*Smith-Vaniz, Collette & Luckhurst, 1999*), suggesting at least some genetic connectivity with other regions (see *Locke et al., 2013*).

Using genetic barcoding of the Cytochrome c oxidase 1 gene (COI), we aim to confirm species identity, assess intraspecific genetic diversity, and determine rates of local and regional genetic connectivity of Bermuda's baitfish populations. Barcoding has proven useful for species verification in families with a high degree of morphological similarity, and has been able to identify new species by integrative taxonomic analysis (*Ward, Hanner & Hebert, 2009*). Furthermore, population genetic analyses can suggest points of origin in mixed populations and provide insights to breeding structures (*Allendorf & Utter, 1979*). Thus, results of this study will provide insights into population stability and can be used to inform future management strategies.

## METHODS

Samples were collected from 10 locations around the islands of Bermuda: the Bermuda Aquarium Museum and Zoo dock in Flatts (BAMZ), Bailey's Bay, Coney Island, Whalebone

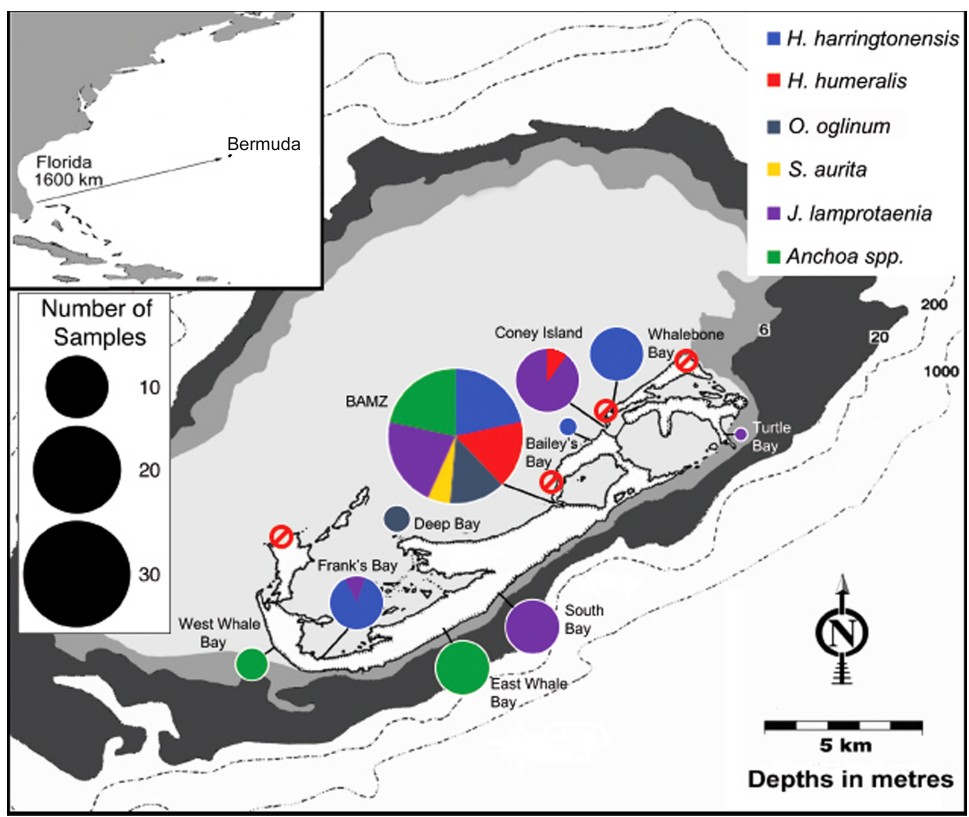

**Figure 1  Map of Bermuda.** Map indicating the locations of bays that are currently closed to net fishing (red circles with strikethrough) and of sampled baitfish populations (pie charts). Size of pie charts represents the total number of individuals sequenced from that location. Colors within the circles represent the relative abundance of each species found at each location. The inset shows the isolated location of Bermuda within the west central Atlantic.

Bay, Turtle Bay, South Bay, East Whale Bay, West Whale Bay, Frank's Bay, and Deep Bay, between July and August 2017 (Fig. 1—clockwise from center). All samples were collected with permission of the Bermuda Government Department of Environment and Natural Resources under special permit SP170303. A total of 111 individuals were collected and visually identified in the field based on previously described gross morphological characteristics (*Smith-Vaniz, Collette & Luckhurst, 1999*). Based on these initial examinations, five species were identified: *Hypoatherina harringtonensis* (Reef silverside), *Anchoa choerostoma* (Bermuda anchovy), *Jenkinsia lamprotaenia* (Dwarf herring), *Harengula humeralis* (Redear herring) and *Sardinella aurita* (Round sardinella). Representative samples of each species from each location were preserved in 95% ethanol for subsequent genetic analyses.

Genomic DNA was extracted from muscle tissue of samples using a Qiagen DNA Blood and Tissue extraction kit following the manufacturer's protocols, resulting in a final volume of 200 µl. The cytochrome c oxidase (COI) gene was amplified from extracted DNA using a primer cocktail developed for fish barcoding as described by *Ivanova et al. (2007)* (COI-3:

C_FishF1t1-C_FishR1t1). All PCRs had a total volume of 12.5 µl and included: 6.25 µl of 5% DMSO, 2.00 µl of $H_2O$, 1.25 µl of 10x Buffer [10 mM KCl, 10 mM $(NH_4)SO_4$, 20 mM Tris–HCl (pH 8.8), 2 mM $MgSO_4$, 0.1% Triton X-100], 0.625 µL $MgCl_2$ (50 mM), 0.125 µl of each primer cocktail, 0.0625 µl of DNTP (10 mM), 0.0625 µl of *Taq* Polymerase (Invitrogen), and 2 µl of DNA template. PCR was optimized at the following: 95 °C for 2 min, 35 cycles of 94 °C for 30 s, 52 °C for 30 s, 72 °C for 1 min, with a final extension at 72 °C for 10 min. PCR products were visualized using a 1.2% agarose gel, with concentration and purity measured using a spectrophotometer. Products were bi-directionally sequenced with universal M13 primers using Sanger Sequencing services provided by Gene Codes Corporation. Resulting sequences were manually edited and aligned using Sequencher® 5.4.6 (Gene Codes Corporation, Ann Arbor, MI, USA) and compared to known sequences in NCBI Blast and GenBank. Accession numbers are listed in Appendix 1, all sequences are available on GenBank (http://www.ncbi.nlm.nih.gov/Genbank). Available sequences of the COI gene for conspecifics from locations outside Bermuda were downloaded from GenBank and used for regional comparisons of genetic structure and connectivity (*Valdez-Moreno et al., 2010*; *Lavoué, Konstantinidis & Chen, 2014*). Sequences from *Anchoa lamprotaenia* (Hildebrand, 1943) from Florida (*Weigt et al., 2012*; *Lavoué, Konstantinidis & Chen, 2014*) were used for comparisons with *A. choerostoma*, as no COI sequences were available for *A. choerostoma*.

Sequences for all species, including those obtained from GenBank, were aligned using MUSCLE (*Edgar, 2004*). Single gene phylogenetic analysis of the genera examined was conducted using the Maximum Likelihood method based on the Tamura-Nei model (*Tamura & Nei, 1993*). Initial trees for the heuristic search were obtained automatically by applying Neighbor-Join and BioNJ algorithms to a matrix of pairwise distances estimated using the Maximum Composite Likelihood (MCL) approach, and then selecting the topology with superior log likelihood value. The analysis involved 154 nucleotide sequences, with a total of 718 positions in the final dataset. Evolutionary analyses were conducted in MEGA X (*Kumar et al., 2018*).

Diversity was assessed within and among locations in Bermuda using standard diversity indices, including number of haplotypes (*Nh*), number of polymorphic sites (*Np*), haplotypic diversity (*h*) (*Nei, 1987*), nucleotide diversity (*pn*) (*Tajima, 1983*; *Nei, 1987*), and mean number of pairwise differences (*pd*) between haplotypes (*Tajima, 1983*) calculated for each species using DnaSP v.5 (*Librado & Rozas, 2009*). Population (location) pairwise $F_{ST}$ (*Hudson, Slatkin & Maddison, 1992*) values, whose significances were assessed through 10,000 permutation tests, were used to calculate differentiation between locations within Bermuda, as well between Bermuda and other regional locations, using ARLEQUIN version 3.5.2 (*Excoffier & Lischer, 2010*). Unweighted analysis of molecular variance (AMOVA; *Excoffier, Smouse & Quattro, 1992*) was also performed to test hierarchical models of genetic variance using pairwise differences among haplotypes as a measure of divergence within and among locations in Bermuda as well as within and among regional locations using ARLEQUIN version 3.5.2 (*Excoffier & Lischer, 2010*).

## RESULTS

A total of 92 individual fish were successfully sequenced (see Appendix 1). Alignments for each species were trivial and required no insertion/deletion events. Sequence identification of 81 individuals confirmed initial identification, while sequence identification did not match the initial field identification for 10 individuals, indicating an error rate of 11% for identification based on gross morphology.

All three samples from Deep Bay, initially identified as *Sardinella aurita*, were molecularly identified as *Opisthonema oglinum*. Five of the seven samples from the BAMZ location that were initially identified as *S. aurita* were also molecularly identified as *O. oglinum*. One sample from Frank's Bay and one sample from Coney Island were identified as *Hypoatherina harringtonensis*, but the resulting sequences matched that of *Jenkinsia lamprotaenia*. COI sequences of individuals morphologically identified as the endemic Bermuda anchovy, *Anchoa choerostoma*, had a 99% identity match (93% query coverage) to the widespread Atlantic species *A. lamprotaenia*. However, no COI sequences were available for *A. choerostoma* on public databases for comparison.

The single gene tree constructed based on maximum likelihood to infer phylogenetic relationships (Fig. 2) indicated that species from the genera *Harengula*, *Opisthonema*, and *Sardinella* are more closely related to each other than to the other species examined, while species from the genera *Jenkinsia* and *Anchoa* are more closely related to each other than to the other species. The genus *Hypoatherina*, in the order Atheriniformes, was the most evolutionarily distant from the other genera.

Haplotype diversity was similar among species based on overlapping standard errors (Table 1). Nucleotide diversity was similar for *A. choerostoma* and *H. harringtonensis*, and for *H. humeralis*, *J. lamprotaenia*, and *O. oglinum*, but was higher for the first two species than for the latter three. The mean number of pairwise differences was highest for *A. choerostoma* at 3.163, decreasing to 0.934 for *J. lamprotaenia*, 0.800 for *H. humeralis*, 0.780 for *H. harringtonensis*, and 0.429 for *O. oglinum*. Diversity of *A. choerostoma* within a given bay was higher at East Whale Bay than at West Whale Bay and BAMZ (Table 1). For *H. harringtonensis*, within-location diversity was higher at BAMZ and Whalebone Bay than at Frank's Bay and Bailey's Bay. Diversity was similar among locations for *H. humeralis* at BAMZ and Coney Island, and among all sampled locations for *J. lamprotaenia*. Likewise, diversity measures did not differ among locations for *O. oglinum* from Deep Bay and BAMZ. While measures of diversity were high for *S. aurita*, the low sample size ($n = 2$) and lack of replicate sites precludes their inclusion in diversity comparisons (Table 1).

Pairwise $F_{ST}$ comparisons between locations within Bermuda were insignificant ($\alpha = 0.05$) for all species examined, indicating no evidence of genetic structure and high levels of genetic connectivity within and among locations (Table 2). An analysis of molecular variance used to test for hierarchical population structure also indicated no significant genetic structure exists among locations in Bermuda for any of the species analyzed, where the majority of variation for all species was found within locations rather than among them (Table 3).

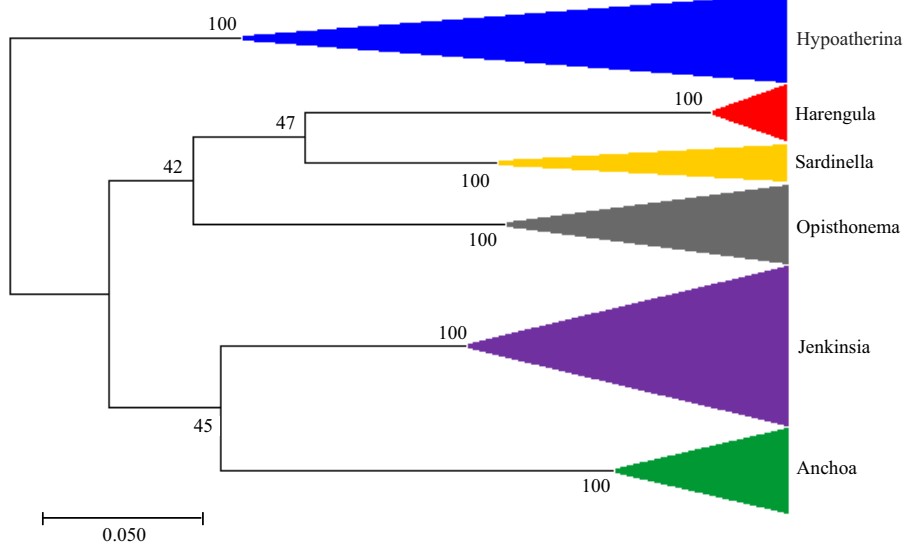

**Figure 2 Molecular Phylogenetic Analysis.** The evolutionary history was inferred from a single gene tree using the Maximum Likelihood method based on the Tamura-Nei model. The tree with the highest log likelihood (−4350.82) is shown. The percentage of trees in which the associated taxa clustered together is shown next to the branches. Initial trees for the heuristic search were obtained automatically by applying Neighbor-Join and BioNJ algorithms to a matrix of pairwise distances estimated using the Maximum Composite Likelihood (MCL) approach, and then selecting the topology with superior log likelihood value. The tree is drawn to scale, with branch lengths in the number of substitutions per site. The analysis involved 154 nucleotide sequences. There were a total of 718 positions in the final dataset. Evolutionary analyses were conducted in MEGA X.

For all species except *H. humeralis*, pairwise $F_{ST}$ comparisons were significant ($\alpha = 0.05$) between populations from Bermuda and those from other regions, indicating strong evidence of genetic structure and limited genetic connectivity across regions (Table 4). An analysis of molecular variance, used to test for hierarchical population structure, also showed significant genetic structure among regions for all species analyzed with the exception of *H. humeralis*, such that the majority of variation was found among regions rather than within them (Table 5).

## DISCUSSION

Based on the COI sequences obtained from baitfish samples in this study, a higher diversity of species was present within the assemblages than was initially recorded based solely on morphological identification in the field at the time of sampling, with 11% of individuals being misidentified. Genetic sequencing revealed the misidentification of several small individuals of *Opisthonema oglinum* that had yet to develop their distinctive threadfin and were thus misidentified as *Sardinella aurita*, and of two small *Jenkinsia lamprotaenia* that had been schooling with *Hypoatherina harringtonensis*. As a result, the total number of species analyzed increased from five to six, and included *Anchoa choerostoma*, *H. harringtonensis*, *H. humeralis*, *J. lamprotaenia*, *O. oglinum*, and *S. aurita*. These results highlight the value of incorporating molecular identification into assessments of species

**Table 1** **Diversity measures.** Standard diversity measures by sampling location for *A. choerostoma*, *H. harringtonensis*, *H. humeralis*, *J. lamprotaenia*, *O. oglinum*, and *S. aurita* including sample size ($n$), number of usable base pairs ($bp$), number of haplotypes ($Nh$), number of polymorphic sites ($Np$), haplotype diversity ($h$), nucleotide diversity ($pn$), and the mean number of pairwise differences ($pd$).

| Species | Location | n | bp | Nh | Np | h | pn | pd |
|---|---|---|---|---|---|---|---|---|
| A. choerostoma | ALL | 20 | 710 | 11 | 16 | 0.763 +/− 0.103 | 0.0045 +/− 0.0010 | 3.163 |
| | West Whale Bay | 4 | 710 | 2 | 1 | 0.500 +/− 0.265 | 0.0007 +/− 0.0004 | 0.500 |
| | East Whale Bay | 8 | 710 | 8 | 12 | 1.00 +/− 0.0039 | 0.0062 +/− 0.0010 | 4.429 |
| | BAMZ | 8 | 710 | 3 | 8 | 0.464 +/− 0.040 | 0.0043 +/− 0.0018 | 3.071 |
| H. harringtonensis | ALL | 25 | 718 | 9 | 8 | 0.640 +/− 0.107 | 0.00295 +/− 0.000002 | 0.780 |
| | Whalebone Bay | 8 | 718 | 4 | 3 | 0.750 +/− 0.139 | 0.0013 +/− 0.0016 | 0.929 |
| | Frank's Bay | 7 | 718 | 2 | 1 | 0.286 +/− 0.196 | 0.0006 +/− 0.0006 | 0.286 |
| | Bailey's Bay | 2 | 718 | 1 | 0 | 0 | 0.00 | 0 |
| | BAMZ | 8 | 718 | 6 | 5 | 0.893 +/− 0.111 | 0.0027 +/− 0.0016 | 1.25 |
| H. humeralis | ALL | 10 | 707 | 5 | 4 | 0.667 +/− 0.163 | 0.0011 +/− 0.0004 | 0.800 |
| | BAMZ | 6 | 707 | 4 | 3 | 0.800 +/− 0.172 | 0.0014 +/− 0.0004 | 1.000 |
| | Coney Island | 4 | 707 | 2 | 1 | 0.500 +/− 0.265 | 0.0007 +/− 0.0004 | 0.500 |
| J. lamprotaenia | ALL | 27 | 718 | 10 | 9 | 0.650 +/− 0.103 | 0.0013 +/− 0.0003 | 0.934 |
| | South Bay | 8 | 718 | 3 | 3 | 0.607 +/− 0.164 | 0.0013 +/− 0.0005 | 0.929 |
| | Coney Island | 9 | 718 | 5 | 4 | 0.722 +/− 0.159 | 0.0015 +/− 0.0005 | 1.056 |
| | Frank's Bay | 1 | 718 | 1 | 0 | 0 | 0 | 0 |
| | Turtle Bay | 1 | 718 | 1 | 0 | 0 | 0 | 0 |
| | BAMZ | 8 | 718 | 5 | 4 | 0.786 +/− 0.151 | 0.0014 +/− 0.0004 | 1.000 |
| O. oglinum | ALL | 8 | 709 | 2 | 1 | 0.429 +/− 0.169 | 0.0006 +/− 0.0002 | 0.429 |
| | Deep Bay | 3 | 709 | 2 | 1 | 0.667 +/− 0.314 | 0.0009 +/− 0.0004 | 0.667 |
| | BAMZ | 5 | 709 | 2 | 1 | 0.400 +/− 0.237 | 0.0006 +/− 0.0003 | 0.400 |
| S. aurita | BAMZ | 2 | 681 | 2 | 3 | 1.000 +/− 0.500 | 0.0044 +/− 0.0022 | 3.000 |

assemblages for morphologically similar species, as diversity may be underestimated when based only on gross morphology (*Zemlak et al., 2009*; *Hubert et al., 2012*). The relationship among the limited sampling of six genera of baitfishes examined in this study was analyzed using a single gene tree based on a Maximum Likelihood phylogenetic approach and broadly follows established relationships (e.g., in *Lavoué, Konstantinidis & Chen, 2014*) in that the genera within the Family Clupeidae (*Harengula, Opisthonema,* and *Sardinella*) were found to be more closely related to each other than to the other genera examined (Fig. 2). However, while our analysis found *Jenkinsia* (Family Dussumieriidae) and *Anchoa* (Family Engraulidae) to be more closely related to each other than to the species in the Clupeidae, recent molecular phylogenies using multiple markers have inferred conflicting sister relationships between the Dussumieriidae, the Engraulidae and the rest of the Clupeoidei (see *Lavoué et al., 2017* and *Egan et al., 2018*). Deciphering these relationships is beyond the scope of the present single gene analysis of five species but, as COI sequences become available for more species, these data may contribute to future analyses to further elucidate the complex evolutionary history of this group of fishes. All of the aforementioned genera belong to the Order Clupeiformes, while *Hypoatherina* belongs to the Order Atheriniformes.

**Table 2** **Genetic connectivity within Bermuda.** Pairwise $F_{ST}$ values among sampling locations for each species. Significant comparisons are indicated in bold ($\alpha = 0.05$).

| *A. choerostoma* | West Whale Bay | East Whale Bay | BAMZ | | |
|---|---|---|---|---|---|
| West Whale Bay | 0 | | | | |
| East Whale Bay | 0.07556 | 0 | | | |
| BAMZ | 0.01604 | −0.05263 | 0 | | |
| *H. harringtonensis* | Whalebone Bay | Frank's Bay | Bailey's Bay | BAMZ | |
| Whalebone Bay | 0 | | | | |
| Frank's Bay | 0.04465 | 0 | | | |
| Bailey's Bay | −0.24551 | −0.3125 | 0 | | |
| BAMZ | −0.02521 | −0.01165 | −0.31765 | 0 | |
| *H. humeralis* | Coney Island | BAMZ | | | |
| Coney Island | 0 | | | | |
| BAMZ | −0.0297 | 0 | | | |
| *J. lamprotaenia* | South Bay | Coney Island | Frank's Bay | Turtle Bay | BAMZ |
| South Bay | 0 | | | | |
| Coney Island | 0.00697 | 0 | | | |
| Frank's Bay | −0.85714 | −0.9 | 0 | | |
| Turtle Bay | −0.85714 | −0.9 | 0 | 0 | |
| BAMZ | 0.00461 | 0.02589 | −1 | −1 | 0 |
| *O. oglinum* | Deep Bay Beach | BAMZ | | | |
| Deep Bay | 0 | | | | |
| BAMZ | −0.29921 | 0 | | | |

Accordingly, *H. harringtonensis* was found to be the most evolutionarily distant from the other genera examined.

Interestingly, the COI sequence results for the endemic Bermuda Anchovy, *A. choerostoma*, were within 1% similarity to the more broadly distributed Big-eye anchovy, *A. lamprotaenia*. *A. choerostoma* was described by Goode in 1874 based on morphological variations from congeneric species. Morphologically, it is most similar to *A. lamprotaenia*, *A. januaria*, *A. cubana* and *A. parva* (*Smith-Vaniz, Collette & Luckhurst, 1999*; *Nizinski & Munroe, 2002*). However, molecular phylogenies have found it most closely related to *A. mitchilli* using the ITS1 region (Johnson 2003 in *Smith-Vaniz & Collette, 2013*), and to *A. hepsetus* using a combination of the genes 12s, 16s, RAG1 and RAG2 (*Li & Ortí, 2007*), but these studies did not include *A. lamprotaenia* . Of these species, *A. hepsetus* is readily distinguished from the others by its longer maxilla (*Nizinski & Munroe, 2002*). *A. choerostoma* is distinguished from *A. mitchilli* by the relative positions of the dorsal fin and anal fin (such that the anal fin in *A. choerostoma* is posterior to the dorsal, whereas the origins of these fins are vertically aligned in *A. mitchilli*); from *A. lamprotaenia* by having a greater number of lower gill rakers (23–30 as opposed to 17–21); and from the remaining similar congenerics by having a notably smaller axillary scale above the pectoral fin (*Smith-Vaniz, Collette & Luckhurst, 1999*; *Nizinski & Munroe, 2002*).

Cytochrome c oxidase is an enzyme in the respiratory chain that catalyzes the conversion of oxygen to water, a critical survival process. Encoded inside mitochondria,

**Table 3 AMOVA within Bermuda.** Analysis of molecular variance within and among sampled locations around Bermuda (Arlequin 3.5.2). Samples from all locations for a given species were considered as a single group. Significant $F_{ST}$ indices are indicated in bold ($\alpha = 0.05$).

| Species | Source of Variation | d.f. | Sum of Squares | Variance components | Percentage of total variation | Fixation indices |
|---|---|---|---|---|---|---|
| *A. choerostoma* | | | | | | |
| | Among locations | 2 | 3.05 | −0.00988 Va | −0.63 | $F_{ST}$: −0.00626 |
| | Within locations | 17 | 27.00 | 1.58824 Vb | 100.63 | |
| | Total | 19 | 30.05 | 1.57835 | | |
| *H. harringtonensis* | | | | | | |
| | Among locations | 3 | 0.88 | −0.01880 Va | −4.88 | $F_{ST}$: −0.04882 |
| | Within locations | 21 | 8.48 | 0.40391 Vb | 104.88 | |
| | Total | 24 | 9.36 | 0.38511 | | |
| *H. humeralis* | | | | | | |
| | Among locations | 1 | 0.35 | −0.01172 Va | −2.97 | $F_{ST}$: −0.02970 |
| | Within loclations | 8 | 3.25 | 0.40625 Vb | 102.97 | |
| | Total | 9 | 3.60 | 0.39453 | | |
| *J. lamprotaenia* | | | | | | |
| | Among locations | 4 | 1.18 | −0.04269 Va | −9.36 | $F_{ST}$: −0.09361 |
| | Within locations | 22 | 10.97 | 0.49874 Vb | 109.36 | |
| | Total | 16 | 12.15 | 0.45605 | | |
| *O. oglinum* | | | | | | |
| | Among locations | 1 | 0.03 | −0.05630 Va | −29.92 | $F_{ST}$: −0.29921 |
| | Within locations | 6 | 1.47 | 0.24444 Vb | 129.92 | |
| | Total | 7 | 1.50 | 0.18815 | | |

the Cytochrome c oxidase subunit 1 gene (COI) is highly conserved among all respiring organisms and is therefore not subject to selective pressures that induce mutation (*Mick, Fox & Rehling, 2011*). It is possible, therefore, that the COI gene may not provide high enough resolution to distinguish between these closely related species within the genus *Anchoa*. To further evaluate and obtain an accurate assessment of endemism, multiple genes should be incorporated and compared among several congeneric species. Coupled with morphological variation, detailed phylogenetic analyses could provide further insights into the classification of the Bermuda anchovy.

Of the locations examined around Bermuda, BAMZ had the highest species diversity, with all six species found at this location (Fig. 1). Conversely, at several locations, only a single species was found. However, this may reflect sampling effort rather than actual diversity, as these sites were sampled less frequently than BAMZ. Yet, BAMZ is also the most centrally located site on the more protected northern shore of Bermuda, and may, therefore, represent an area of species accumulation (*Tittensor et al., 2010*), at least for inshore species, that results in higher baitfish diversity. Among the species, *J. lamprotaenia* was the most widely distributed, being found at five of the 10 locations, followed by *H. harringtonensis* at four of the 10 locations. Given the high estimates of connectivity among locations for all six species examined, however, distributions are likely wider than reflected by the somewhat limited sampling effort in this study.

**Table 4  Regional genetic connectivity.** Pairwise $F_{ST}$ values among regions for each species. Significant comparisons are indicated in bold ($\alpha = 0.05$).

| *A. choerostoma* | n | Bermuda | Florida | |
|---|---|---|---|---|
| Bermuda | 20 | 0 | | |
| Florida (*A. lamprotaenia*) | 5 | **0.46211** | **0** | |
| *H. harringtonensis* | n | Bermuda | Belize | |
| Bermuda | 25 | 0 | | |
| Belize | 2 | **0.97053** | **0** | |
| *H. humeralis* | n | Bermuda | Mexico | |
| Bermuda | 10 | 0 | | |
| Mexico | 4 | 0.08142 | 0 | |
| Belize | 2 | −0.32353 | −0.26316 | 0 |
| *J. lamprotaenia* | n | Bermuda | Mexico | Belize |
| Bermuda | 27 | 0 | | |
| Mexico | 17 | **0.98389** | 0 | |
| Belize | 5 | **0.92381** | **0.92381** | 0 |
| *O. oglinum* | n | Bermuda | Mexico | Brazil |
| Bermuda | 8 | 0 | | |
| Mexico | 4 | **0.56408** | 0 | |
| Brazil | 6 | **0.69100** | 0.15109 | 0 |
| *S. aurita* | n | Bermuda | Turkey | Israel |
| Bermuda | 2 | 0 | | |
| Turkey | 3 | **0.68902** | 0 | |
| Israel | 6 | **0.80840** | −0.05882 | 0 |

Estimates of genetic diversity within and among baitfish assemblages in Bermuda indicate high degrees of mixing between locations for all six species examined. For these small-bodied species, this mixing likely occurs predominantly during the larval phase (*Schultz, 2000*; *Lavoué, Konstantinidis & Chen, 2014*), but may also occur during later life stages as a result of short- or long-term movements between locations that may be driven by food availability, predator density, reproductive cycles or adverse conditions (*Hugie & Dill, 1994*; *Olsson et al., 2006*; *Udyawer et al., 2013*; *Currey et al., 2015*). These high rates of local connectivity mean that management should consider Bermuda's baitfish species as single, highly mixed populations. As such, the distribution of bays that are closed to net fishing in order to protect them is of less importance than it might be if subregional genetic differentiation had been detected, and there is no immediate need to close additional bays in central or western parishes, or along the south shore, in order to maintain local genetic diversity.

On a larger scale, most marine species occurring in Bermuda are restricted to the Tropical Northwestern Atlantic biogeographic province (*Spalding et al., 2007*; *Locke et al., 2013*), which is largely equivalent to the Caribbean biogeographic province of *Kulbicki et al. (2013)*, and the baitfish species examined here generally follow this distributional pattern. In contrast, *S. aurita* is found on both sides of the Atlantic Ocean and into the Mediterranean Sea (*Aquamaps, 2016*). However, range size does not necessarily reflect

**Table 5 AMOVA among regions.** Analysis of molecular variance among and within regions (Arlequin 3.5.2). Populations from all regions for a given species were considered as a single group. Significant $F_{ST}$ indices are indicated in bold ($\alpha = 0.05$).

| Species | Source of Variation | d.f. | Sum of Squares | Variance components | Percentage of total variation | Fixation indices |
|---|---|---|---|---|---|---|
| *A. choerostoma / lamprotaenia* | | | | | | |
| | Among regions | 1 | 9.67 | 1.05522 Va | 46.21 | $F_{ST}$: **0.46211** |
| | Within regions | 23 | 28.25 | 1.22826 Vb | 53.79 | |
| | Total | 14 | 37.92 | 2.28348 | | |
| *H. harringtonensis* | | | | | | |
| | Among regions | 1 | 46.05 | 12.33171 Va | 97.05 | $F_{ST}$: **0.97053** |
| | Within regions | 25 | 9.36 | 0.37440 Vb | 2.95 | |
| | Total | 26 | 55.41 | 12.70611 | | |
| *H. humeralis* | | | | | | |
| | Among regions | 2 | 0.90 | −0.01357 Va | −2.75 | $F_{ST}$: −0.02747 |
| | Within regions | 13 | 6.60 | 0.50769 Vb | 102.75 | |
| | Total | 15 | 7.50 | 0.49412 | | |
| *J. lamprotaenia* | | | | | | |
| | Among regions | 2 | 1061.95 | 38.14984 Va | 94.25 | $F_{ST}$: **0.94252** |
| | Within regions | 46 | 107.03 | 2.32675 Vb | 5.75 | |
| | Total | 48 | 1168.98 | 40.47659 | | |
| *O. oglinum* | | | | | | |
| | Among regions | 2 | 14.53 | 1.09471 Va | 53.83 | $F_{ST}$: **0.53831** |
| | Within regions | 15 | 14.08 | 0.93889 Vb | 46.17 | |
| | Total | 17 | 28.61 | 2.0336 | | |
| *S. aurita* | | | | | | |
| | Among regions | 2 | 8.23 | 1.08507 Va | 65.86 | $F_{ST}$: **0.65859** |
| | Within regions | 8 | 4.50 | 0.56250 Vb | 34.14 | |
| | Total | 10 | 12.73 | 1.64757 | | |

modern dispersal patterns (*Lester et al., 2007*) and, for marine fishes, small body size is correlated with lower dispersal capacity and increased rates of endemism (*DeMartini & Friedlander, 2004*; *Bradbury et al., 2008*). This seems likely to apply to baitfish species, with implications for regional connectivity for an isolated island like Bermuda.

The present study documents significant genetic differentiation between conspecific populations of baitfishes in Bermuda and those in other regions (Table 4; $F_{ST}$; $p < 0.05$), and significant divergence among regions (Table 5; AMOVA; $p < 0.05$) for *J. lamprotaenia*, *H. harringtonensis*, *O. oglinum*, and *S. aurita*, as well as between congeneric populations of *Anchoa sp.* among regions. These results suggest that gene flow, and therefore exchange of individuals, is limited between populations of these species in Bermuda and those in other regions. Populations of *H. humeralis*, on the other hand, show no evidence of genetic structure, suggesting broad genetic connectivity exists across the Caribbean / Tropical West Atlantic for this species. Importantly, these results have major implications for management as strong genetic divergence in *J. lamprotaenia*, *H. harringtonensis*, *O. oglinum*, and *S. aurita* suggests that populations of these species in Bermuda are likely self-seeding and locally

maintained. Thus, local declines in baitfish abundances are not likely to be replenished from distant populations.

## CONCLUSION

Baitfishes depend on shallow inshore areas, which are capable of sustaining great diversity and densities of organisms (*Nagelkerken et al., 2001*; *Ray & Carleton Ray, 2005*; *Vasconcelos et al., 2011*; *Araújo et al., 2017*), but are also extensively modified and threatened by human activities such as overfishing, pollution, coastal development and habitat degradation, which may impact fish communities (*Kennish, 2002*; *Sax & Gaines, 2003*; *Ribeiro et al., 2008*; *Johnston & Roberts, 2009*; *Ecoutin et al., 2010*; *Last et al., 2010*; *Araújo et al., 2017*). The limited genetic connectivity of baitfish populations among Western Atlantic regions documented here indicates restricted influx of new individuals to Bermuda and highlights the vulnerability of local populations to natural and anthropogenic perturbations. As such, it is important to monitor both fish communities and environmental parameters in Bermuda's nearshore habitats (*Araújo et al., 2017*), and to adapt management measures accordingly, in order to conserve these ecologically and economically important species.

## ACKNOWLEDGEMENTS

We are grateful to Jirani Welch, Chris Flook and Robbie Smith for their assistance with collecting samples.

### Funding

This study was funded by the UK DEFRA Darwin Initiative (grant DPLUS064) and the National Science Foundation REU Program (NSF-REU OCE-1757475), and supported by the Bermuda Zoological Society, the Bermuda Institute for Ocean Sciences and the Bermuda Government Department of Environment and Natural Resources. The funders had no role in study design, data collection and analysis, decision to publish, or preparation of the manuscript.

### Grant Disclosures

The following grant information was disclosed by the authors:
UK DEFRA Darwin Initiative: DPLUS064.
National Science Foundation REU Program: NSF-REU OCE-1757475.
Bermuda Zoological Society, the Bermuda Institute for Ocean Sciences.
Bermuda Government Department of Environment and Natural Resources.

### Competing Interests

The authors declare there are no competing interests.

## Author Contributions

- Gretchen Goodbody-Gringley conceived and designed the experiments, performed the experiments, analyzed the data, contributed reagents/materials/analysis tools, prepared figures and/or tables, authored or reviewed drafts of the paper, approved the final draft.
- Emma Strand performed the experiments, analyzed the data, prepared figures and/or tables, authored or reviewed drafts of the paper, approved the final draft.
- Joanna M. Pitt conceived and designed the experiments, contributed reagents/materials/analysis tools, authored or reviewed drafts of the paper, approved the final draft.

## Animal Ethics

The following information was supplied relating to ethical approvals (i.e., approving body and any reference numbers):

All samples were collected with permission of the Bermuda Government Department of Environment and Natural Resources under special permit SP170303.

## Field Study Permissions

The following information was supplied relating to field study approvals (i.e., approving body and any reference numbers):

Field collections were approved by the Bermuda Government, Department of Environment and Natural Resources (SP170303).

## Data Availability

Data are available at NCBI GenBank via accession numbers MK871561 to MK871655. Accession numbers are also available in Appendix 1.

## Supplemental Information

Supplemental information for this article can be found online at http://dx.doi.org/10.7717/peerj.7244#supplemental-information.

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
