# Peer review of "Molecular characterization of nearshore baitfish populations in Bermuda to inform management"

_PeerJ, doi:10.7717/peerj.7244_

## Round 0.1 · original submission · Minor Revisions

We have now heard back from 2 expert referees, both of whom are supportive of your work being acceptable for publication pending some minor revisions. They are largely straightforward and should not be difficult to revise. The only comment of substance is the objection by one referee to your description of the shortcomings of morphological characters compared to the molecular data. They point out that these species can be easily distinguished using morphological characters by a taxonomic specialist at any developmental stage. They would like the wording of this section softened for accuracy, but this should not be difficult.

Given the small number of relatively minor edits requested by the referees, provided you address them adequately in the revision and outline the changes to the manuscript in your rebuttal letter, I do not expect the paper will need additional review, and I look forward to seeing the revised manuscript.

Reviewer 1 ·

Basic reporting

This paper is clearly written and largely error-free. The introduction and discussion mostly appropriately review the literature, although I have a few suggested changes/additions (see general comments below).

Experimental design

Methods were clearly and comprehensively explained. Research is original and fits into the aims and scope of this journal. Molecular methods were not cutting edge, but sufficient to explore the questions posed by this study.

Validity of the findings

I think this study’s findings are valid and conclusions and are clearly stated (but see comments below for a few specific suggestions).

Additional comments

I really appreciated this study and thee clear writing and high quality condition of this manuscript! It is great to see such economically and ecologically important fishes getting some much needed attention!

Comment 1: You mention misidentification of some individuals (Line 227). Can you comment on whether or not these were small individuals and thus harder to identify?

Comment 2: The Lavoue et al. 2014 reference is somewhat outdated in the context in which it appears on Line 237. If you are going to discuss clupeiform phylogenetics it might be more appropriate to reference this paper, which is more comprehensive and up to date: Egan et al. (2018) Phylogenetic analysis of trophic niche evolution reveals a latitudinal herbivory gradient in Clupeoidei (herrings, anchovies, and allies).

Comment 3: You state that your findings about the relationships between Anchoa and Jenkinsia are congruent with recent research (line 236). This is not the case, although this is likely due to taxon sampling and the small quantity of genetic data used in this study. Contradictory results are reported in Lavoué et al. 2013 & 2014 references in your manuscript and several other recent studies including these:

Egan et al. (2018) Phylogenetic analysis of trophic niche evolution reveals a latitudinal herbivory gradient in Clupeoidei (herrings, anchovies, and allies).

Bloom DD & Lovejoy NR (2014) The evolutionary origins of diadromy inferred from a time-calibrated phylogeny for Clupeiformes (herring and allies).

Comment 4: Wording in Line 248 is not quite right in my opinion. It would be more appropriate to say something like “phylogenetic analyses based-upon morphology suggest it Is more closely related to….”

Comment 5: The paragraph starting at line 289 needs to be revised. I am not quite sure what the authors are trying to get across. Please add a topic sentence and edit to make the take home message of this paragraph clear.

Comment 5: It is not clear to me how the statement that “baitfish families typically have global distributions” relates to the findings of this study (line 289).

Comment 6: The claims in lines 292-295 need additional citation and need to be elaborated upon because right now they are an oversimplification. Lavoué et al. 2013 simply study the biogeographic history of Clupeoidei and do not compare rates of baitfish endemism in tropical versus temperate regions. Furthermore, there are many small, tropical clupeoidei with very large distributions (e.g. Encrasicholina spp.) as is also the case with some other tropical baitfish species (e.g. Atherinidae spp.). It might be worthwhile to point out some specific examples of endemism (e.g. Stolephorus nelsonii or Nematalosa vlaminghi) or reference some studies describing rates of endemism in small marine fishes to make your points more convincing.

·

Basic reporting

The article "Molecular characterization of nearshore baitfish populations in Bermuda to inform management" by Goodbody-Gringley et al. meets all the standards of the journal.
The work shows the importance of baitfish for management, not only as a food source for larger predators higher up the food chain but also to protect species in isolated areas.

Experimental design

The experimental design appears to be very good.
However, I am insisting on changing some of the wordings. On page 17 at the beginning of the discussion the authors mention the difficulty to identify two of the target species (O. oglinum and S. aurita). Although this is a possibility with other species, this is not the case with O. oglinum and S. aurita because the meristics of the dorsal and anal fins are clearly different with no overlap between the two species. Even before the threadfin develops in O. oglinum they should easily be distinguishable. Even the larval stages have different pigment patterns. I strongly recommend to rephrase this section (see comment in pdf)

Validity of the findings

No comment

Additional comments

I really enjoyed the paper and believe it is of high importance for the management and therefore protection of commercially less important species/baitfishes.

---

## Round 0.2 · accepted · Accept

I have now had a chance to read through your revised manuscript, and am satisfied with your responses to the referee comments. Therefore, I am happy to move your manuscript forward into production.